# OpenReview forum: "GhostEI-Bench: Do Mobile Agent Resilience to Environmental Injection in Dynamic On-Device Environments?"
_ICLR.cc/2026/Conference — ICLR 2026 Poster_

### Official Review · Reviewer_i5H1 · 2025-10-22

**Soundness:** 3
**Presentation:** 4
**Contribution:** 3
**Rating:** 8
**Confidence:** 3

**Summary:**

This paper introduces GhostEI-Bench a benchmark for examining how vulnerable Vision-Language Models (VLMs) powering mobile agents are to environmental injection. Environmental injection attacks manipulate an agent’s visual perception by embedding deceptive UI elements like overlays or spoofed notifications directly into the interface. These attacks can bypass language safeguards and can cause severe harms, including privacy breaches or device compromise. The benchmark uses Android emulators to simulate realistic workflows and employs a judge LLM to pinpoint failure points in the agents’ perception, recognition, or reasoning. The authors perform experiments on 7 current state-of-the-art models and show they are highly susceptible to these attacks, underscoring the urgent need for more robust and secure vision-language agents.

**Strengths:**

- The paper is well written
- The paper is addressing a pressing issue
- Well crafted benchmarks can be very useful to other researchers
- Well presented results
- Interesting results looking at "self refection" and "reasoning"

**Weaknesses:**

- "Pop up" based attacks are not novel
- The novelty over works like Chen et al. 2025 seems very limited.
@article{chen2025evaluating,
  title={Evaluating the Robustness of Multimodal Agents Against Active Environmental Injection Attacks},
  author={Chen, Yurun and Hu, Xavier and Yin, Keting and Li, Juncheng and Zhang, Shengyu},
  journal={arXiv preprint arXiv:2502.13053},
  year={2025}
}
- Reliance on LLM as a judge without any evaluation of the judges effectiveness or other methods of evaluation
- 112 test cases is not that many
- Deceptive Instruction seem like a strange choice to include in a benchmark for Environmental Injection as these do not come from the Environment
- no error bars or stand deviations, granted running repeats here would likely be costly
- Missing references to Adversarial Image Attacks On GUI agents such as

@article{fu2024imprompter, title={Imprompter: Tricking llm agents into improper tool use}, author={Fu, Xiaohan and Li, Shuheng and Wang, Zihan and Liu, Yihao and Gupta, Rajesh K and Berg-Kirkpatrick, Taylor and Fernandes, Earlence}, journal={arXiv preprint arXiv:2410.14923}, year={2024} }

@article{aichberger2025attacking, title={Attacking multimodal os agents with malicious image patches}, author={Aichberger, Lukas and Paren, Alasdair and Gal, Yarin and Torr, Philip and Bibi, Adel}, journal={arXiv preprint arXiv:2503.10809}, year={2025} }

@article{wu2024dissecting, title={Dissecting adversarial robustness of multimodal lm agents}, author={Wu, Chen Henry and Shah, Rishi and Koh, Jing Yu and Salakhutdinov, Ruslan and Fried, Daniel and Raghunathan, Aditi}, journal={arXiv preprint arXiv:2406.12814}, year={2024} }

@article{wang2025manipulating, title={Manipulating Multimodal Agents via Cross-Modal Prompt Injection}, author={Wang, Le and Ying, Zonghao and Zhang, Tianyuan and Liang, Siyuan and Hu, Shengshan and Zhang, Mingchuan and Liu, Aishan and Liu, Xianglong}, journal={arXiv preprint arXiv:2504.14348}, year={2025} }

**Questions:**

- Why do you consider Deceptive Instruction and a Environmental Injection?

- What are the main differences between your work and prior work looking at Environmental Injection attacks in GUI agents?

---

> ### Author Response · Authors · 2025-11-21
> **Response to Reviewer Reviewer i5H1 (1/3)**
>
> We thank Reviewer i5H1 for the well positive assessment of our paper's writing, relevance, and the potential utility of our benchmarks. We appreciate the detailed feedback and the suggestion of relevant literature. Below, we address the weaknesses and questions raised.
>
> ### **On Weakness 1 & 2 & Question 2: Novelty and Distinction from Concurrent Work**
>
> We sincerely thank the reviewer for pointing out this important related work. We acknowledge that Chen et al. (2025)[1] is a significant contribution that effectively establishes the feasibility and severity of active injection attacks. GhostEI-Bench **offers contributions beyond such foundational work** by providing a **systematic evaluation standard** with a distinct focus on quantifying diverse risk consequences.
>
> 1. **From Feasibility to Standardization:** While prior work demonstrated that such attacks are possible, GhostEI-Bench provides the tooling (Judge, Metrics like VR) to rigorously measure how vulnerable different models are.
> 2. **Systematic Risk Assessment:** GhostEI-Bench **contributes a distinct evaluation framework** focused on a structured taxonomy covering **7 critical risk fields** (e.g., Fraud, Privacy Leakage) across **14 real-world apps**. Our evaluation **specifically emphasizes** characterizing the **specific risk consequences**, enabling a granular analysis of how environmental injections lead to distinct harms (e.g., distinguishing between benign failure and attack success).
> 3. **Comprehensive Threat Model:** Our benchmark integrates three distinct attack vectors (Deceptive Instruction, Static, and Dynamic Injection) to holistically measure the "safety surface" of mobile agents. This provides a multi-dimensional view of agent robustness.
> 4. **Concurrent Contribution:** As cited in our original paper (Section 2.2), we consider Chen et al. (2025) as a critical concurrent work. In our revision, we will further expand the discussion in the Related Work section to explicitly delineate how GhostEI-Bench’s focus on benchmarking metrics (VR, FAS/PAS) and risk diversity serves.
>
> [1] Chen, Yurun, et al. "Evaluating the robustness of multimodal agents against active environmental injection attacks." *Proceedings of the 33rd ACM International Conference on Multimedia*. 2025.

---

> ### Author Response · Authors · 2025-11-21
> **Response to Reviewer Reviewer i5H1 (2/3)**
>
> ### **On Weakness 3 & 6: Evaluation Reliability and Statistical Rigor**.
>
> We fully agree that the reliability of our evaluation metrics is crucial.
>
> 1. **LLM Judge Validation (We Add New Human-LLM Agreement Study):** Consistent with prevalent methodologies in the GUI agent community (e.g., DigiRL[2], DistRL[3]), we employ the LLM-as-a-judge paradigm. To address concerns about the judge's effectiveness, we conducted a new agreement study. We tasked a human expert with independently re-annotating the execution trajectories generated by GPT-4o in MobileAgent v2 across the benchmark to benchmark the automated judge's accuracy. We analyzed the discrepancies between the evaluations generated by GPT-4o judge and those conducted manually. Among the 110 cases examined, **only 7 showed inconsistencies**, resulting in a consistency rate of **93.6%**. The comparison of the full distribution of outcomes (TC/FAS/PAS/BF/VR%) is shown below:
>
>
>     | Judge | TC | FAS | PAS | BF | VR (%) |
>     | --- | --- | --- | --- | --- | --- |
>     | **GPT-4o** | 38 (34.5%) | 33 (30.0%) | 12 (10.9%) | 28 (25.5%) | **54.88** |
>     | **Human Expert** | 35 (31.8%) | 34 (30.9%) | 10 (9.9%) | 32 (29.1%) | **56.41** |
>     | *Difference* | *-2.7%* | *+0.9%* | *-1.0%* | *+3.6%* | *+1.53* |
>
>     The results show exceptional alignment between the automated judge and human annotations. These few mismatches primarily arise from ambiguous boundary definitions for harmful queries (e.g., subjective nuances in classifying a user question) rather than model hallucinations. This high consistency confirms that our methodology is statistically reliable and appropriate for assessing agent robustness.
>
> 2. **Statistical Significance:**
>
>     We appreciate the suggestion regarding statistical rigor. To address this, we conducted repeated evaluations (3 independent runs) on representative models to quantify the variance.
>
>     | Model | TC (%) | FAS (%) | PAS (%) | BF (%) | VR (%) |
>     | --- | --- | --- | --- | --- | --- |
>     | Gemini-2.5 Pro | 47.90 ± 1.85 | 25.17 ± 0.47 | 7.30 ± 0.73 | 20.30 ± 1.85 | 40.70 ± 0.50 |
>     | GPT-4o | 34.53 ± 0.78 | 30.3 ± 1.85 | 10.0 ± 0.73 | 25.76 ± 1.16 | 54.25 ± 2.24 |
>
>     The results demonstrate low standard deviations. This stability confirms that the observed performance gaps between models are statistically significant and robust to execution noise, validating the reliability of our benchmark metrics.
>
>
> [2] Bai, Hao, et al. "Digirl: Training in-the-wild device-control agents with autonomous reinforcement learning." *Advances in Neural Information Processing Systems* 37 (2024): 12461-12495.
>
> [3] Wang, Taiyi, et al. "Distrl: An asynchronous distributed reinforcement learning framework for on-device control agents." *arXiv preprint arXiv:2410.14803* (2024).
>
> ### **On Weakness 4: Limited Number of Test Cases**
>
> We appreciate the your comment on scale. We clarify that GhostEI-Bench prioritizes the **fidelity** and **systematic density** of dynamic attacks over the raw quantity typical of static benchmarks.
>
> 1. **High-Fidelity Dynamic Execution:** Unlike large-scale datasets generated automatically which often suffer from logical inconsistencies or triviality, every single scenario in GhostEI-Bench was meticulously hand-crafted and verified by security experts. This ensures that each attack vector is logically sound, contextually realistic, and executable in a dynamic environment.
> 2.  **Systematic Design Space Coverage:** We maximized statistical diversity through rigorous curation rather than random sampling. The benchmark is structured to span **7 representative domains** (e.g., Finance, Social Media) and **7 critical risk fields** (e.g., Fraud, Privacy Leakage). Crucially, **each test case is designed to be multi-dimensional**, often mapping to multiple risk fields simultaneously (as shown in Figure 3). This efficient design allows us to comprehensively cover the critical "risk surface" of mobile agents within a compact, high-quality dataset.

---

> ### Author Response · Authors · 2025-11-21
> **Response to Reviewer Reviewer i5H1 (3/3)**
>
> ### **On Weakness 5 & Question 1: Inclusion of Deceptive Instructions**
>
> We include Deceptive Instruction as part of GhostEI-Bench not to treat it as an environmental attack, but to ensure the **completeness and interpretability** of the overall safety evaluation. Our benchmark is built on a unified threat model (Sec. 3.1; Fig. 2) that spans three complementary risk sources: (i) deceptive user instructions, (ii) static environmental risks, and (iii) dynamic environmental injections.
>
> In this design, Deceptive Instruction serves as a **baseline safety-alignment assessment**. It allows us to **decouple** an agent’s *intrinsic* alignment vulnerabilities (failure to refuse harmful prompts) from its *extrinsic* susceptibility to environmental manipulations. More specifically, to assert that an agent was fooled by an Environmental Injection (e.g., an Overlay), we must first know if the agent would have refused the harmful request had it been plain text. An agent that accepts a plain-text harmful prompt lacks basic safety alignment; thus, refusing the text while accepting an overlay allows us to isolate vulnerabilities specifically attributable to the dynamic environment. Thus, Deceptive Instruction is crucial for the internal validity of our Environmental Injection claims.
>
> ### **On Weakness 7: Missing Related Works**
>
> We appreciate your suggestion regarding adversarial image attacks on GUI agents. Incorporating this broader context of adversarial vision attacks will indeed strengthen the paper. In the revised version, we will expand the Related Work section to contrast these pixel-level or adversarial attacks with our dynamic environmental injection attacks.

---

> ### Author Response · Authors · 2025-11-28
> **Looking forward to further feedback**
>
> Dear Reviewer i5H1,
>
> We hope this message finds you well.
>
> **As the rebuttal period deadline is approaching**, we would greatly appreciate it if you could kindly acknowledge and respond to our rebuttal for Submission 8451. We are looking forward to receiving any further questions or suggestions for improvement. If you find our responses satisfactory, we appreciate it if you could kindly revise your rating based on your updated assessment of our work.
>
> We sincerely look forward to your further comments. Thank you very much for your support.
>
> Best regards,
> Authors of Submission 8451

---

### Official Review · Reviewer_qhur · 2025-10-30

**Soundness:** 1
**Presentation:** 3
**Contribution:** 2
**Rating:** 4
**Confidence:** 4

**Summary:**

This paper introduces GhostEI-Bench, the first comprehensive benchmark designed to evaluate mobile agent robustness against environmental injection attacks in dynamic, on-device environments. The authors construct 110 test cases across 7 application domains and evaluate 8 state-of-the-art VLMs, revealing vulnerability rates between 40-55% for most models.

**Strengths:**

- **Comprehensive Benchmark Design**: The benchmark is well-structured with 110 carefully curated test cases spanning 7 domains (Communication, Finance, Social Media, etc.) and 7 critical risk fields (Fraud, Privacy Leakage, etc.), providing systematic coverage of the threat landscape.
- The hook-based trigger mechanism for dynamic injection is technically sound and allows precise timing of adversarial UI elements during agent execution.

**Weaknesses:**

- Limited Novelty and Insight in Attack Methods: While the benchmark itself is valuable, the attack techniques employed (such as overlays and pop-ups) are not novel. It appears the authors are incorporating a recently proposed attack method into their evaluation framework. The claim that their attack is distinct from those used in text-based environments is questionable. For dynamic environments, where malicious prompts or misleading text in images are used, the attacks seem conceptually similar. The authors should provide further explanation or experiments to clarify the fundamental differences between these attack methods.

- **Inconsistencies in results across tables.**: *Table 1 (p.7)* reports **GPT-5-chat-latest** with **PAS = 5 (4.6%)**, but *Table 2 (p.9)* lists the “no reflection” variant at **PAS = 35 (31.8%)**. Similarly, *Table 1* lists **Claude-3.7-Sonnet** with **PAS = 13 (11.8%)**, while *Table 3 (p.9)* reports **PAS = 43 (39.1%)** for the base model. Are these results from different subsets, runs, or judge prompts? What accounts for these significant discrepancies?


- **Judge reliability.**: The evaluation heavily relies on the LLM judge, but there is no analysis of its accuracy or consistency in the authors' setup. This lack of validation could introduce potential biases and inconsistencies, undermining the reliability of the results.

**Questions:**

Have you explored any defensive strategies against these attacks? For example, could prompting agents to be more cautious about overlays or implementing visual consistency checks reduce vulnerability? Alternatively, would using a separate LLM to audit agent actions at each step be helpful?

---

> ### Author Response · Authors · 2025-11-21
> **Response to Reviewer qhur (1/2)**
>
> We thank Reviewer qhur for the constructive feedback and for acknowledging the comprehensive design and technical soundness of GhostEI-Bench. We address the concerns regarding novelty, data consistency, and evaluator reliability below.
>
> ### **On Weakness 1: Limited Novelty of Attack Methods**
>
> We respectfully disagree that our dynamic injections are conceptually identical to static text/image attacks. We agree that primitives like overlays are not new in isolation. However, we respectfully clarify that GhostEI-Bench’s contribution lies in formalizing **Dynamic Environmental Injection** as a distinct *threat vector for agents*, which differs fundamentally from text-based or static prompt attacks:
>
> 1. **Temporal & Contextual Dependency:** Unlike static prompt injections (which exist before execution), our attacks are strictly time-sensitive. An overlay is only effective if injected at the precise moment of user interaction (handled by our hook mechanism). This tests the agent's *realtime* decision-making stability, not just static safety alignment.
> 2. **Multi-Modal Conflict:** In text attacks, the conflict is within the prompt. In our case, the conflict is between a benign textual instruction and a deceptive visual state. This requires the agent to resolve cross-modal discrepancies, a capability that static benchmarks cannot evaluate.
> 3. **Systematic Risk Quantification:** As noted in Section 3.1, we move beyond task failure to measure tangible harms (Fraud, Privacy Leakage) across 7 critical risk fields. With meticulously designed cases, GhostEI-Bench provides the first reproducible framework to quantify these specific vulnerabilities.
>
> As shown in Figure 4, "Attack Vector Vulnerabilities", agents like GPT-4o have reasonable refusal rates for Static Attack, yet fail significantly more often against Dynamic Injection. If the attacks were conceptually similar, the failure rates would be correlated; instead, we see a distinct vulnerability profile for dynamic attacks, which is an indirect and empirical evidence for our attacking novelty methods.
>
> ### **On Weakness 2: Result Inconsistencies Across Tables**
>
> We sincerely apologize for this confusion. It is correct that the Partial Attack Success (PAS) values for the baseline models in Tables 2 and 3 did not match the values reported in our main results in Table 1.
>
> This discrepancy stems from a clerical error rooted in an early stage of our data analysis.  During the early phase of the work, we had an internal discussion on the most appropriate way to define PAS: specifically, whether Full Attack Success (FAS) should be counted as a subset of PAS (i.e., making PAS an umbrella term for any successful attack). The numbers in Tables 2 and 3 were inadvertently drawn from this preliminary analysis, where we had calculated `PAS_Reported = PAS_Actual + FAS`.
>
> For the submitted version of paper, we decided on a stricter, more fine-grained definition where FAS and PAS are reported separately to better distinguish the severity of different attack outcomes. The results in Table 1 correctly reflect this final, more precise definition. The inconsistency between tables stems from the failure to update all data synchronously.
>
> We take **GPT-5-chat-latest** as an example to illustrate this point:
>
> 1. The incorrect PAS of 35 (31.8%) in Table 2 was the sum of its FAS count (30) and its true PAS count (5).
> 2. The calculation is: 35 (reported in old Table 2) - 30 (FAS count from Table 1) = 5 (the correct PAS count). This matches the 5 (4.6%) value accurately reported in Table 1.
> 3. The same logic applies to **Claude-3.7-Sonnet**: 43 (reported in old Table 3) - 30 (FAS count) = 13 (the correct PAS count).
>
> We have meticulously re-verified all data and have now **corrected Tables 2 and 3** in the revised paper to align with our definitive evaluation criteria, ensuring complete consistency across all tables. Thanks once again for helping us significantly improve the accuracy and clarity of our paper.

---

> ### Author Response · Authors · 2025-11-21
> **Response to Reviewer qhur (2/2)**
>
> ### **On Weakness 3: Reliability of the LLM Judge**
>
> We fully agree that the reliability of our LLM Judge is paramount to the validity of our results.Consistent with prevalent methodologies in the GUI agent community (e.g., DigiRL[1], DistRL[2]), we employ the LLM-as-a-judge paradigm.  To further verify the reliability of the LLM judge, we have conducted a new Human–LLM Agreement study on the execution trajectories generated by GPT-4o in MobileAgent v2 across the benchmark.  The same trajectories, initially scored by the automated GPT-4o judge, are re-annotated by a human expert, and we compare the full distribution of outcomes (TC/FAS/PAS/BF/VR%). We analyzed the discrepancies between the evaluations generated by GPT-4o judge and those conducted manually.
>
> | Judge | TC | FAS | PAS | BF | VR (%) |
> | --- | --- | --- | --- | --- | --- |
> | **GPT-4o** | 38 (34.5%) | 33 (30.0%) | 12 (10.9%) | 28 (25.5%) | **54.88** |
> | **Human Expert** | 35 (31.8%) | 34 (30.9%) | 10 (9.9%) | 32 (29.1%) | **56.41** |
> | *Difference* | *-2.7%* | *+0.9%* | *-1.0%* | *+3.6%* | *+1.53* |
>
> The results show exceptional alignment between the automated judge and human annotations. These few mismatches primarily arise from ambiguous boundary definitions for harmful queries (e.g., whether a user question should be treated as harmful or benign) rather than model hallucinations. This high consistency confirms that our methodology, i.e., using GPT-4o as a judge, is statistically reliable and appropriate for assessing agent robustness at scale.
>
> [1] Bai, Hao, et al. "Digirl: Training in-the-wild device-control agents with autonomous reinforcement learning." *Advances in Neural Information Processing Systems* 37 (2024): 12461-12495.
>
> [2] Wang, Taiyi, et al. "Distrl: An asynchronous distributed reinforcement learning framework for on-device control agents." *arXiv preprint arXiv:2410.14803* (2024).
>
> ### **On Question: Defensive Strategies against these Attacks**
>
> Yes, we have explored several strategies in Section 4.4 including whether reasoning modes or reflection models can effectively address these threats, while the primary contribution of our paper lies in formalizing and quantifying the problem.
> The question has further motivated us to design a new experiment that directly evaluates these defense strategies.
>
> 1. **On "Cautious Prompting(CP)" :**
>     - To test its efficacy, we have designed a new experiment.
>
>
>         | Model | TC ↑ | FAS ↓ | PAS ↓ | BF ↓ | VR (%) ↓ |
>         | --- | --- | --- | --- | --- | --- |
>         | gpt-4o (w/o CP) | 38 (34.5%) | 33 (30.0%) | **12 (10.9%)** | **28 (25.5%)** | 54.88 |
>         | gpt-4o (w/ CP) | **42 (38.2%)** | **21 (19.1%)** | 17 (15.5%) | 30 (27.3%) | **47.50** |
>     - **Analysis:** The results show that explicit safety prompts provide a measurable defense. Cautious Prompting reduced the Vulnerability Rate from **54.88% to 47.50%**. Notably, this improvement is primarily driven by a significant drop in **Full Attack Success (30.0%** to **19.1%)**, while **Partial Attack Success** increased (10.9% to 15.5%). This suggests that while textual constraints make agents more hesitant to complete fully malicious workflows (shifting failures from FAS to PAS), they are insufficient to completely override the visual allure of deceptive UI elements. Interestingly, Task Completion (TC) also slightly improved, indicating that safety constraints do not necessarily compromise utility.
> 2. **On "Using a separate LLM to audit agent actions at each step":**
>
>     We sincerely appreciate this insight, as it perfectly aligns with the **"Self-Reflection" mechanism** we evaluated in **Section 4.4**.
>     In our implementation (based on the Mobile-Agent-v2 framework), the reflection module essentially acts as the "auditor" you suggested—it reviews the agent's planned actions against the current state to detect errors or anomalies before execution.
>
>     Our results (presented in **Table 2**) empirically confirm your hypothesis: this "auditing" strategy is indeed effective. For instance, enabling reflection for **GPT-5 reduced its Vulnerability Rate (VR) from 35.71% to 26.58%**, and for **GPT-4o from 54.87% to 48.75%**. This demonstrates that a separate auditing step can help the agent identify inconsistencies between the user's intent and the deceptive environment (e.g., an overlay). However, we also observed a trade-off: strict auditing can sometimes make agents overly cautious, slightly reducing the Task Completion rate for some models (e.g., GPT-4o). We thank you for highlighting this direction.
>
>
> We believe that developing robust, multi-modal defense mechanisms is a critical frontier, and we hope our GhostEI-Bench will catalyze further research in this urgent direction.

---

> ### Author Response · Authors · 2025-11-28
> **Looking forward to further feedback**
>
> Dear Reviewer qhur,
>
> We hope this message finds you well.
>
> **As the rebuttal period deadline is approaching**, we would greatly appreciate it if you could kindly acknowledge and respond to our rebuttal for Submission 8451. We are looking forward to receiving any further questions or suggestions for improvement. If you find our responses satisfactory, we appreciate it if you could kindly revise your rating based on your updated assessment of our work.
>
> We sincerely look forward to your further comments. Thank you very much for your support.
>
> Best regards,
> Authors of Submission 8451

---

### Official Review · Reviewer_XueB · 2025-11-03

**Soundness:** 3
**Presentation:** 3
**Contribution:** 3
**Rating:** 6
**Confidence:** 3

**Summary:**

This paper presents GhostEI-Bench, the first benchmark designed to systematically evaluate the robustness of mobile vision-language agents against environmental injection attacks—a novel threat vector where adversarial UI elements such as overlays and spoofed notifications are injected dynamically into mobile environments. The benchmark executes 110 adversarial scenarios in realistic Android emulators across seven domains and multiple risk categories. It includes an LLM-based judge that performs fine-grained trajectory-level evaluation of agent actions to distinguish between benign failures, task completion, and varying levels of compromise. Empirical evaluations on leading VLM agents reveal widespread vulnerabilities, with vulnerability rates ranging from 16% to over 50%. The paper further analyzes failure patterns, showing that dynamic overlays and disinformation/fraud contexts are the most effective attack modes.

**Strengths:**

1. The paper introduces a novel threat model that integrates dynamic environmental injection attacks while also encompassing traditional prompt-based and static UI manipulations. This dual coverage makes the benchmark both innovative and methodologically complete.
2. The inclusion of an automated LLM judge capable of analyzing full action trajectories and corresponding screenshots enables fine-grained, scalable assessment of perception, recognition, and reasoning failures.
3. The paper is clearly organized and well-written.
4. The benchmark encompasses a diverse set of applications across multiple mobile domains, ensuring that the evaluation results are representative and practically relevant.

**Weaknesses:**

1. The reliability and objectivity of results depend heavily on the LLM evaluator’s accuracy, yet potential biases or inconsistencies of this judge model are not rigorously validated.
2. The study focuses primarily on comparing different VLMs without analyzing how variations in agent frameworks may influence vulnerability to environmental injection.
3. The benchmark evaluation excludes specialized or fine-tuned VLMs (such as UI-Tars), leaving open the question of how domain-specific training affects robustness.
4. The paper mainly concentrates on attack evaluation and vulnerability analysis, with insufficient exploration of potential defense mechanisms or mitigation techniques.
5. Although the benchmark is well designed, it currently includes only 110 scenarios, which may limit the statistical diversity and coverage of real-world attack cases.

**Questions:**

What is the performance of various agent frameworks and agent-task-fine-tuned VLMs on the proposed benchmarks?

---

> ### Author Response · Authors · 2025-11-21
> **Response to Reviewer XueB (1/3)**
>
> We sincerely thank Reviewer XueB for their constructive feedback and for recognizing the novelty of our threat model, the thoroughness of our automated evaluation protocol, and the practical relevance of our diverse mobile application domains. We have conducted extensive new experiments to address the concerns regarding evaluator reliability, framework impact, specialized agents, and defense mechanisms.
>
> ### **On Weakness 1: Reliability of the LLM Judge**
>
> We fully agree that the reliability of our LLM Judge is paramount to the validity of our results.Consistent with prevalent methodologies in the GUI agent community (e.g., DigiRL[1], DistRL[2]), we employ the LLM-as-a-judge paradigm. To further verify the reliability of the LLM judge, we have conducted a new Human–LLM Agreement study on the execution trajectories generated by GPT-4o in MobileAgent v2 across the benchmark.  The same trajectories, initially scored by the automated GPT-4o judge, are re-annotated by a human expert, and we compare the full distribution of outcomes (TC/FAS/PAS/BF/VR%). We analyzed the discrepancies between the evaluations generated by GPT-4o judge and those conducted manually. Among the 110 cases examined, **only 7 showed inconsistencies**, resulting in a consistency rate of **93.6%**.
>
> | Judge | TC | FAS | PAS | BF | VR (%) |
> | --- | --- | --- | --- | --- | --- |
> | **GPT-4o** | 38 (34.5%) | 33 (30.0%) | 12 (10.9%) | 28 (25.5%) | **54.88** |
> | **Human Expert** | 35 (31.8%) | 34 (30.9%) | 10 (9.9%) | 32 (29.1%) | **56.41** |
> | *Difference* | *-2.7%* | *+0.9%* | *-1.0%* | *+3.6%* | *+1.53* |
>
> The results show exceptional alignment between the automated judge and human annotations. These few mismatches primarily arise from ambiguous boundary definitions for harmful queries (e.g., whether a user question should be treated as harmful or benign) rather than model hallucinations.  This high consistency confirms that our methodology, i.e., using GPT-4o as a judge, is statistically reliable and appropriate for assessing agent robustness at scale.
>
> [1] Bai, Hao, et al. "Digirl: Training in-the-wild device-control agents with autonomous reinforcement learning." *Advances in Neural Information Processing Systems* 37 (2024): 12461-12495.
>
> [2] Wang, Taiyi, et al. "Distrl: An asynchronous distributed reinforcement learning framework for on-device control agents." *arXiv preprint arXiv:2410.14803* (2024).
>
> ### **On Weakness 2 & Question: Lack of Framework Analysis**
>
> Thanks for this insightful suggestion. We investigated whether the vulnerability is intrinsic to the VLM or introduced by the agent framework. We compared **Mobile-Agent-v2** (our default) against **AppAgent** [3], using the same VLMs.
>
> | Model | TC ↑ | FAS ↓ | PAS ↓ | BF ↓ | VR (%) ↓ |
> | --- | --- | --- | --- | --- | --- |
> | GPT-4o (AppAgent) | 37 (33.6%) | **24 (21.8%)** | **12 (10.9%)** | 38 (34.5%) | **50.00** |
> | GPT-4o(Mobile Agent v2) | **38 (34.6%)** | 33 (30.0%) | **12 (10.9%)** | **28 (25.5%)** | 54.87 |
> | Qwen2.5-VL-72B-Instruct(AppAgent) | 38 (34.6%) | 27 (24.5%) | **14 (12.7%)** | **32 (29.1%)** | 52.56 |
> | Qwen2.5-VL-72B-Instruct(Mobile Agent v2) | **42 (38.2%)** | **12 (10.9%)** | 17 (15.5%) | 40 (36.4%) | **41.42** |
>
> **Key Findings:**
>
> 1. **Framework Bias:** Agent frameworks can both amplify or reduce a model’s robustness. For instance, Mobile Agent v2 substantially weakens GPT-4o (VR: 54.87%) due to its elevated FAS rate (30.0%), yet it *improves* the robustness of Qwen2.5-VL (VR: 41.42%). This demonstrates that frameworks introduce their own bias and may interact differently with specific VLM architectures.
> 2. **Intrinsic VLM Properties:** Even under the same AppAgent framework, GPT-4o exhibits stronger robustness (VR: 50.00%) than Qwen2.5-VL (VR: 52.56%), driven largely by better resistance to prompt-induced attacks (PAS: 10.9% vs. 12.7%). This indicates that intrinsic VLM security characteristics remain central, though their expression is mediated by the surrounding agent framework.
> 3. **Value of the Benchmark:** Crucially, despite these variations, all combinations of frameworks and VLMs exhibit high Vulnerability Rates (VR consistently > 41%). None are robust against environmental injection. This persistence of failure across different architectures demonstrates that GhostEI-Bench captures a fundamental and universal threat in mobile agents, validating the benchmark’s broad applicability and significance beyond any single system configuration.
>
> [3] Zhang, Chi, et al. "Appagent: Multimodal agents as smartphone users." *Proceedings of the 2025 CHI Conference on Human Factors in Computing Systems*. 2025.

---

> ### Author Response · Authors · 2025-11-21
> **Response to Reviewer XueB (2/3)**
>
> ### **On Weakness 3 & Question: Missing Domain-Specific VLMs**
>
> We sincerely thank you for highlighting the importance of evaluating domain-specific, agent-task-fine-tuned VLMs. In response, we have added new experimental evaluations on the UI-TARS series of agents[4], which are specifically designed and fine-tuned for mobile GUI tasks.
>
> | Model | TC ↑ | FAS ↓ | PAS ↓ | BF ↓ | VR (%) ↓ |
> | --- | --- | --- | --- | --- | --- |
> | UI-TARS-7B-SFT | 29 (26.4%) | 20 (18.2%) | **16 (14.5%)** | 46 (41.8%) | 56.25 |
> | UI-TARS-1.5-7B | **45 (40.9%)** | **19 (17.3%)** | 20 (18.18%) | **27 (24.5%)** | **46.99** |
>
> **Key Findings:**
>
> 1. **Task Completion (TC)**: UI-TARS-1.5-7B shows a substantial increase in Task Completion rate (**40.9%** vs 26.4%) compared to the earlier version, indicating that fine-tuning effectively enhances its core Agent capabilities.
> 2. **Robustness (VR%):** UI-TARS-1.5-7B demonstrates improved robustness, with the overall Vulnerability Rate (VR%) decreasing from 56.25% to 46.99%. Notably, the Benign Failure (BF) rate drops significantly (24.5% vs 41.8%), indicating that the agent is far more capable of interacting with the interface. While this increased agency leads to a rise in Partial Attack Success (PAS)—as the model engages more actively with the environment rather than failing benignly—the simultaneous increase in Task Completion and reduction in overall Vulnerability confirm that the model performance has been comprehensively improved in version 1.5.
> 3. **Behavioral Trade-off (FAS vs. PAS)**:Compared with general-purpose VLMs running under Mobile Agent v2, the specialized training of UI-TARS leads to a distinct failure pattern. When facing dynamic pop-up or overlay attacks, UI-TARS tends to remain more task-oriented, attempting to preserve or restore the original execution trajectory. This behavior results in a higher PAS—the model partially complies with injected instructions—but a lower FAS, as the agent is less likely to be fully diverted into executing the attacker’s intended malicious workflow. This pattern suggests that domain-specific fine-tuning helps the agent stay on task, providing protection against full derailment; however, it simultaneously increases the likelihood of partial inducement or incidental clicks in response to injected UI elements.
> 4. **Persistent Vulnerability:** Similarly, even these domain-specific, fine-tuned experts maintain a high Vulnerability Rate (>46%). This indicates that current GUI-focused training improves usability but fails to instill safety awareness against dynamic injections. This validates GhostEI-Bench as a challenging and necessary benchmark that exposes security gaps even in the most specialized models.
>
> [4] Qin, Yujia, et al. "Ui-tars: Pioneering automated gui interaction with native agents." *arXiv preprint arXiv:2501.12326* (2025).
>
> ### **On Weakness 4: Limited Discussion of Defenses**
>
> We respectfully claim that we we had already taken initial steps to investigate this topic in Section 4.4 (Effects of Reflection and Reasoning Mechanisms), and we agree it is better to explore defense mechanisms more deeply.
>
> Specifically, as presented in Section 4.4, we included an evaluation of self-reflection modules as a built-in defense mechanism. Our findings in Table 2 demonstrate that reflection enables agents to re-assess their actions and can reduce vulnerability to environmental injections by identifying inconsistencies between user instructions and adversarial UI elements.
>
> Inspired by your valuable feedback, we have conducted a new, supplementary experiment to further strengthen this aspect of our work. This new experiment specifically investigates the impact of using **Cautious Prompting, which is inspired by the Reviewer qhur,** explicitly instructing the agent to prioritize system stability and other risks, as a direct defensive measure.
>
> | Model | TC ↑ | FAS ↓ | PAS ↓ | BF ↓ | VR (%) ↓ |
> | --- | --- | --- | --- | --- | --- |
> | gpt-4o (w/o CP) | 38 (34.5%) | 33 (30.0%) | **12 (10.9%)** | **28 (25.5%)** | 54.88 |
> | gpt-4o (w/ CP) | **42 (38.2%)** | **21 (19.1%)** | 17 (15.5%) | 30 (27.3%) | **47.50** |
>
> Our results show that Cautious Prompting provides a measurable layer of defense. It effectively reduced the Vulnerability Rate of GPT-4o from 54.88% to 47.50%. Notably, this improvement is driven by a significant drop in Full Attack Success (30.0% to 19.1%), while Partial Attack Success increased (10.9% to 15.5%). This indicates that Cautious Prompting makes agents more hesitant to execute fully malicious workflows, effectively downgrading critical compromises into partial failures. However, the persistence of partial success suggests that textual constraints alone are insufficient to completely override the visual allure of high-fidelity environmental deception.

---

> ### Author Response · Authors · 2025-11-21
> **Response to Reviewer XueB (3/3)**
>
> ### **On Weakness 5: Limited Scenario Diversity**
>
> While 110 seems numerically smaller than static QA datasets, we emphasize that GhostEI-Bench prioritizes the fidelity and systematic density of dynamic attacks over the raw scale typical of static benchmarks.
>
> - **High-Quality Expert Curation:** Unlike large-scale datasets generated automatically which often suffer from logical inconsistencies or triviality, every single scenario in GhostEI-Bench was meticulously hand-crafted and verified by security experts. This ensures that each attack vector is logically sound, contextually realistic, and executable in a dynamic environment.
> - **Dynamic Complexity:** GhostEI-Bench is not a static QA dataset; it requires maintaining valid states across 110 multi-step execution trajectories with real time hook injections. The engineering effort to guarantee high fidelity for each handcrafted case is significantly higher than static image generation.
> - **Systematic Design Space Coverage:** We maximized statistical diversity through rigorous curation rather than random sampling. The benchmark is structured to span 7 representative domains (e.g., Finance, Social Media) and 7 critical risk fields (e.g., Fraud, Privacy Leakage). Crucially, each test case is designed to be multi-dimensional, often mapping to multiple risk fields simultaneously (as shown in Figure 3). This efficient design allows us to comprehensively cover the critical "risk surface" of mobile agents within a compact, high quality dataset.
> - **Statistical Significance of Vulnerability:** Our results show consistent, high vulnerability rates (40%-55%) across 8 state-of-the-art VLMs(including GPT-4o and Gemini-2.5). This demonstrates that the current benchmark size is statistically sufficient to expose systematic failures in state-of-the-art models without needing thousands of redundant samples.

---

> ### Author Response · Authors · 2025-11-28
> **Looking forward to further feedback**
>
> Dear Reviewer XueB,
>
> We hope this message finds you well.
>
> **As the rebuttal period deadline is approaching**, we would greatly appreciate it if you could kindly acknowledge and respond to our rebuttal for Submission 8451. We are looking forward to receiving any further questions or suggestions for improvement. If you find our responses satisfactory, we appreciate it if you could kindly revise your rating based on your updated assessment of our work.
>
> We sincerely look forward to your further comments. Thank you very much for your support.
>
> Best regards, Authors of Submission 8451

---

### Author Response · Authors · 2025-11-21
**General Response to Reviewers**

We sincerely thank all reviewers (XueB, qhur, i5H1) for their constructive feedback and encouraging assessment of **GhostEI-Bench**. We are encouraged that the reviews consistently recognize: **(i) the critical importance and practical relevance of studying environmental injection risks for mobile agents; (ii) the comprehensive structure of GhostEI-Bench covering diverse domains and risk fields; and (iii) the value of a well-crafted benchmark in systematically evaluating agent robustness against these emerging threats.**

**We further conduct experiments and try to handle the main concerns of all reviews:**

1. **Reliability of LLM judge (XueB, qhur, i5H1):** To address validation concerns, we conducted a Human-LLM agreement study and repeated independent runs, confirming high expert alignment and statistical stability.
2. **Impact of Frameworks and Specialized Models (XueB):** We added experiments with AppAgent and UI-TARS; results confirm that high vulnerability rates persist across different frameworks and specialized models despite fine-tuning.
3. **Exploration of Defense Mechanisms (XueB, qhur):** We tested "Cautious Prompting" defenses; findings indicate that while full attack success drops, agents remain susceptible to partial compromise due to visual deception.
4. **Novelty of Contribution and Scope Validity (qhur, i5H1):** We clarified our contribution is quantifying real-world risks, and justified Deceptive Instructions as a baseline to isolate visual injection vulnerabilities from intrinsic alignment failures.

---

### Author Response · Authors · 2025-12-03
**Rebuttal Summary (for AC and SAC)**

**Dear AC and SACs,**

We fully understand that, due to the rare technical issues, you are facing a significantly higher workload than usual. We remain sincerely grateful for the additional time and care you are investing, and we greatly appreciate your efforts in keeping the review process running smoothly under these challenging circumstances.

We also want to express our sincere thanks to all reviewers ($\color{brown}{\textbf{XueB}}$, $\color{purple}{\textbf{qhur}}$, $\color{green}{\textbf{i5H1}}$) for their constructive feedback and encouraging assessment. We are particularly encouraged that the reviewers unanimously recognize the high value of our contribution, specifically highlighting:

- **Importance of the Environmental injection (**Reviewer $\color{brown}{\textbf{XueB}}$,  $\color{green}{\textbf{i5H1}}$**):** The critical necessity of formalizing Environmental Injection as a distinct and pressing threat vector for mobile agents.
- **Comprehensive Design (Reviewer** $\color{brown}{\textbf{XueB}}$, $\color{purple}{\textbf{qhur}}$**):** The systematic structure of our benchmark, which ensures broad and realistic coverage across 7 representative domains and multiple critical risk fields.
- **Technical Soundness (Reviewer** $\color{purple}{\textbf{qhur}}$, $\color{green}{\textbf{i5H1}}$**):** The robustness of our technical implementation (e.g., the hook-based trigger mechanism) and the utility of the fine-grained LLM-based evaluation protocol for future research.
- **Clarity & Presentation (**Reviewer $\color{green}{\textbf{i5H1}}$, $\color{brown}{\textbf{XueB}}$**):** The high quality of writing and clear organization of the paper, which is highlighted as "excellent" and "clearly organized and well-written."
- **Insightful Analysis (Reviewer** $\color{green}{\textbf{i5H1}}$**):** The value of our empirical findings beyond the dataset itself, particularly the "interesting results looking at self-reflection and reasoning".

To assist your final assessment, we would like to clarify the status of our rebuttal. It is **unfortunate** that the reviewers are not able to acknowledge our responses or update their ratings, particularly given that they raised **no new questions** during former process. Specifically, **we have incorporated comprehensive revisions to address all of the reviewers' initial comments, as detailed below:**

- $\color{blue}{\textbf{Abstract (Page 1) and Section 4.1 (Page 7)}}$: We expand our evaluation scope to include diverse agent frameworks (e.g., AppAgent) and specialize GUI-finetuned models (e.g., UI-TARS) to explicitly assess how framework variations and domain-specific training affect robustness. ($\color{brown}{\textbf{XueB}}$)
- $\color{blue}{\textbf{Section 2.1 (Page 2)}}$: We update the Related Work to include the latest "reasoning-oriented RL approaches" (e.g., UI-R1, GUI-R1, InfiGUI-R1) to substantiate the validity of the LLM-as-a-judge paradigm, demonstrating that model-based evaluation is a popular method in cutting-edge research. ($\color{brown}{\textbf{XueB}}$, $\color{purple}{\textbf{qhur}}$, $\color{green}{\textbf{i5H1}}$)
- $\color{blue}{\textbf{Section 2.2 (Page 3)}}$: We add a subsection "Adversarial Vulnerabilities in Multimodal Agents" and include missing references (e.g., Imprompter, Attacking multimodal OS agents, etc.) to contextualize our work better. ($\color{green}{\textbf{i5H1}}$)
- $\color{blue}{\textbf{Section 4.1 (Page 7)}}$: We expand the experimental setup about the AppAgent framework and UI-TARS models to analyze if framework variations and domain-specific fine-tuning affect robustness. ($\color{brown}{\textbf{XueB}}$)
- $\color{blue}{\textbf{Table 1 (Page 8)}}$: We expand the main results table to include more evaluations of different agent frameworks (Mobile-Agent-v2 vs. AppAgent) and specialized GUI-finetuned models (UI-TARS series). ($\color{brown}{\textbf{XueB}}$) We also correct the data inconsistencies previously found between tables to ensure accurate reporting. ($\color{purple}{\textbf{qhur}}$)
- $\color{blue}{\textbf{Appendix D.2 (Page 20)}}$: We add a experiment of "Cautious Prompting" as a defense approach to evaluate whether textual safety prompts can effectively prevent environmental injections. ($\color{brown}{\textbf{XueB}}$, $\color{purple}{\textbf{qhur}}$)
- $\color{blue}{\textbf{Appendix D.3 (Page 21)}}$: We add a "Statistical Significance Analysis" section, performing repeated evaluations to report standard deviations and ensure the robustness of our findings against execution noise. ($\color{green}{\textbf{i5H1}}$)

**Lastly,** we sincerely thank all of you. Your guidance and the reviewers' feedback have significantly helped us refine and improve this work.

---

### Meta-Review · Area_Chair_u4Fh · 2026-01-06

**Summary:**

This work proposes GhostEI-Bench, a benchmark designed to measure the robustness of mobile agents against environmental injection attacks. The topic is timely and of clear interest, and the benchmark is generally well designed. Reviewers appreciated the authors’ responses to earlier concerns, particularly the validation of the LLM-based evaluation protocol and the inclusion of experiments with a broader range of VLMs.

Some concerns remain regarding the claimed novelty of the pop-up–based attacks and the overall diversity of tasks, which were not fully resolved in the current version (see `Reviewer Concerns` for more details). In addition, the experimental setup assumes relatively broad execution privileges for agents, whereas real-world mobile systems often involve multiple layers of user permissions, potentially limiting realism.

I encourage the authors to think carefully about how to reflect the comments or resolve the questions from reviewers in the camera-ready version.

**Reviewer Concerns:**

### Reviewer XueB
* [resolved] Potential issues with LLM evaluators: Correlation between LLM-based evaluation and human evaluation was clarified.
* [resolved] Adding specialized or fine-tuned VLMs: The UI-TARS series of agents was added.
* [resolved] Lack of investigation into defense methods: Cautious Prompting was tested.
* [partially resolved] Task diversity: Task diversity is inherently subjective. However, evaluating the method under a wider range of variations (e.g., through LLM-based augmentation) would strengthen the empirical validation.

### Reviewer qhur

* [resolved] Judge reliability: Addressed similarly to Reviewer XueB.
* [remaining] Limited novelty and insight in attack methods: While the authors argue that the agentic setup makes the attacks more dynamic, this justification is not entirely convincing. Fundamentally, the proposed attacks do not appear to deviate substantially from previously studied pop-up and overlay–based attacks, as noted by the reviewers. Moreover, execution-based attacks have been explored in prior work, such as multi-turn conversational settings. The multi-modal aspect is also not entirely novel, making the concern about limited novelty valid.
* [resolved] Inconsistencies in results across tables: The authors clarified that the inconsistencies were due to reporting mistakes in the original submission, which is somewhat concerning but now resolved.


### Reviewer i5H1
* [partially resolved] Task diversity: see Reviewer XueB
* [remaining] Limited novelty in attack methods: see Reviewer qhur
* [remaining] Novelty over Chen et al. (2025)
* [resolved] LLM-based evaluation
* [resolved] Missing error bars / standard deviations
* [remaining] Main differences between this work and prior studies on environmental injection

**Reviewer Scores:**

* Reviewer XueB: 6 $\rightarrow$ 6

* Reviewer qhur: 4 $\rightarrow$ 4

* Reviewer i5H1: 8 $\rightarrow$ (6 or 8)

---

### Decision · Program_Chairs · 2026-01-26

Accept (Poster)